# A Safe Screening Rule for Sparse Logistic Regression

**Jie Wang**
Arizona State University
Tempe, AZ 85287
jie.wang.ustc@asu.edu

**Jiayu Zhou**
Arizona State University
Tempe, AZ 85287
jiayu.zhou@asu.edu

**Jun Liu**
SAS Institute Inc.
Cary, NC 27513
jun.liu@sas.com

**Peter Wonka**
Arizona State University
Tempe, AZ 85287
peter.wonka@asu.edu

**Jieping Ye**
Arizona State University
Tempe, AZ 85287
jieping.ye@asu.edu

## Abstract

The $\ell_1$-regularized logistic regression (or sparse logistic regression) is a widely used method for simultaneous classification and feature selection. Although many recent efforts have been devoted to its efficient implementation, its application to high dimensional data still poses significant challenges. In this paper, we present a fast and effective **s**parse **lo**gistic **re**gression **s**creening rule (Slores) to identify the "0" components in the solution vector, which may lead to a substantial reduction in the number of features to be entered to the optimization. An appealing feature of Slores is that the data set needs to be scanned only once to run the screening and its computational cost is negligible compared to that of solving the sparse logistic regression problem. Moreover, Slores is independent of solvers for sparse logistic regression, thus Slores can be integrated with any existing solver to improve the efficiency. We have evaluated Slores using high-dimensional data sets from different applications. Experiments demonstrate that Slores outperforms the existing state-of-the-art screening rules and the efficiency of solving sparse logistic regression can be improved by one magnitude.

## 1   Introduction

Logistic regression (LR) is a popular and well established classification method that has been widely used in many domains such as machine learning [4, 7], text mining [3, 8], image processing [9, 15], bioinformatics [1, 13, 19, 27, 28], medical and social sciences [2, 17] etc. When the number of feature variables is large compared to the number of training samples, logistic regression is prone to over-fitting. To reduce over-fitting, regularization has been shown to be a promising approach. Typical examples include $\ell_2$ and $\ell_1$ regularization. Although $\ell_1$ regularized LR is more challenging to solve compared to $\ell_2$ regularized LR, it has received much attention in the last few years and the interest in it is growing [20, 25, 28] due to the increasing prevalence of high-dimensional data. The most appealing property of $\ell_1$ regularized LR is the sparsity of the resulting models, which is equivalent to feature selection.

In the past few years, many algorithms have been proposed to efficiently solve the $\ell_1$ regularized LR [5, 12, 11, 18]. However, for large-scale problems, solving the $\ell_1$ regularized LR with higher accuracy remains challenging. One promising solution is by "screening", that is, we first identify the "*inactive*" features, which have $0$ coefficients in the solution and then discard them from the optimization. This would result in a reduced feature matrix and substantial savings in computational cost and memory size. In [6], El Ghaoui *et al.* proposed novel screening rules, called "SAFE", to accelerate the optimization for a class of $\ell_1$ regularized problems, including LASSO [21], $\ell_1$

regularized LR and $\ell_1$ regularized support vector machines. Inspired by SAFE, Tibshirani *et al.* [22] proposed "strong rules" for a large class of $\ell_1$ regularized problems, including LASSO, elastic net, $\ell_1$ regularized LR and more general convex problems. In [26], Xiang et al. proposed "DOME" rules to further improve SAFE rules for LASSO based on the observation that SAFE rules can be understood as a special case of the general "sphere test". Although both strong rules and the sphere tests are more effective in discarding features than SAFE for solving LASSO, it is worthwhile to mention that strong rules may mistakenly discard features that have non-zero coefficients in the solution and the sphere tests are not easy to be generalized to handle the $\ell_1$ regularized LR. To the best of our knowledge, the SAFE rule is the only screening test for the $\ell_1$ regularized LR that is "safe", that is, it only discards features that are guaranteed to be absent from the resulting models.

In this paper, we develop novel screening rules, called "Slores", for the $\ell_1$ regularized LR. The proposed screening tests detect inactive features by estimating an upper bound of the inner product between each feature vector and the "dual optimal solution" of the $\ell_1$ regularized LR, which is unknown. The more accurate the estimation is, the more inactive features can be detected. An accurate estimation of such an upper bound turns out to be quite challenging. Indeed most of the key ideas/insights behind existing "safe" screening rules for LASSO heavily rely on the least square loss, which are not applicable for the $\ell_1$ regularized LR case due to the presence of the logistic loss. To this end, we propose a novel framework to accurately estimate an upper bound. Our key technical contribution is to formulate the estimation of an upper

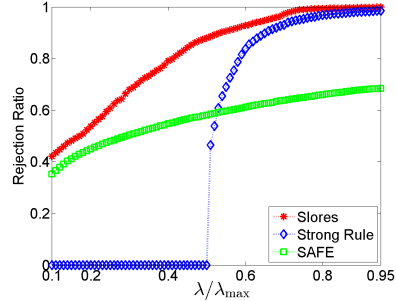

Figure 1: Comparison of Slores, strong rule and SAFE on the prostate cancer data set.

bound of the inner product as a constrained convex optimization problem and show that it admits a closed form solution. Therefore, the estimation of the inner product can be computed efficiently. Our extensive experiments have shown that Slores discards far more features than SAFE yet requires much less computational efforts. In contrast with strong rules, Slores is "safe", i.e., it never discards features which have non-zero coefficients in the solution.

To illustrate the effectiveness of Slores, we compare Slores, strong rule and SAFE on a data set of prostate cancer along a sequence of 86 parameters equally spaced on the $\lambda/\lambda_{max}$ scale from 0.1 to 0.95, where $\lambda$ is the parameter for the $\ell_1$ penalty and $\lambda_{max}$ is the smallest tuning parameter [10] such that the solution of the $\ell_1$ regularized LR is 0 [please refer to Eq. (1)]. The data matrix contains 132 patients with 15154 features. To measure the performance of different screening rules, we compute the rejection ratio which is the ratio between the number of features discarded by screening rules and the number of features with 0 coefficients in the solution. Therefore, the larger the rejection ratio is, the more effective the screening rule is. The results are shown in Fig. 1. We can see that Slores discards far more features than SAFE especially when $\lambda/\lambda_{max}$ is large while the strong rule is not applicable when $\lambda/\lambda_{max} \leq 0.5$. We present more results and discussions to demonstrate the effectiveness of Slores in Section 6. For proofs of the lemmas, corollaries, and theorems, please refer to the long version of this paper [24].

## 2    Basics and Motivations

In this section, we briefly review the basics of the $\ell_1$ regularized LR and then motivate the general screening rules via the KKT conditions. Suppose we are given a set of training samples $\{\mathbf{x}_i\}_{i=1}^m$ and the associate labels $\mathbf{b} \in \Re^m$, where $\mathbf{x}_i \in \Re^p$ and $b_i \in \{1, -1\}$ for all $i \in \{1, \dots, m\}$. The $\ell_1$ regularized logistic regression is:

$$\min_{\beta, c}\ \frac{1}{m} \sum_{i=1}^m \log(1 + \exp(-\langle\beta, \bar{\mathbf{x}}_i\rangle - b_i c)) + \lambda\|\beta\|_1, \qquad \text{(LRP}_\lambda\text{)}$$

where $\beta \in \Re^p$ and $c \in \Re$ are the model parameters to be estimated, $\bar{\mathbf{x}}_i = b_i \mathbf{x}_i$, and $\lambda > 0$ is the tuning parameter. We denote by $\overline{\mathbf{X}} \in \Re^{m \times p}$ the data matrix with the $i^{th}$ row being $\bar{\mathbf{x}}_i$ and the $j^{th}$ column being $\bar{\mathbf{x}}^j$.

Let $\mathcal{C} = \{\theta \in \Re^m : \theta_i \in (0,1), i = 1, \dots, m\}$ and $f(y) = y \log(y) + (1-y) \log(1-y)$ for $y \in (0,1)$. The dual problem of (LRP$_\lambda$) [24] is given by

$$\min_{\theta} \left\{ g(\theta) = \frac{1}{m} \sum_{i=1}^{m} f(\theta_i) : \ \|\bar{\mathbf{X}}^T \theta\|_\infty \leq m\lambda, \langle \theta, \mathbf{b} \rangle = 0, \theta \in \mathcal{C} \right\}. \qquad \text{(LRD}_\lambda\text{)}$$

To simplify notations, we denote the feasible set of problem (LRD$_\lambda$) as $\mathcal{F}_\lambda$, and let $(\beta_\lambda^*, c_\lambda^*)$ and $\theta_\lambda^*$ be the optimal solutions of problems (LRP$_\lambda$) and (LRD$_\lambda$) respectively. In [10], the authors have shown that for some special choice of the tuning parameter $\lambda$, both of (LRP$_\lambda$) and (LRD$_\lambda$) have closed form solutions. In fact, let $\mathcal{P} = \{i : b_i = 1\}$, $\mathcal{N} = \{i : b_i = -1\}$, and $m^+$ and $m^-$ be the cardinalities of $\mathcal{P}$ and $\mathcal{N}$ respectively. We define

$$\lambda_{max} = \tfrac{1}{m}\|\bar{\mathbf{X}}^T \theta_{\lambda_{max}}^*\|_\infty, \qquad (1)$$

where

$$[\theta_{\lambda_{max}}^*]_i = \begin{cases} \frac{m^-}{m}, & \text{if } i \in \mathcal{P}, \\ \frac{m^+}{m}, & \text{if } i \in \mathcal{N}, \end{cases} \quad i = 1, \dots, m. \qquad (2)$$

($[\cdot]_i$ denotes the $i^{th}$ component of a vector.) Then, it is known [10] that $\beta_\lambda^* = 0$ and $\theta_\lambda^* = \theta_{\lambda_{max}}^*$ whenever $\lambda \geq \lambda_{max}$. When $\lambda \in (0, \lambda_{max}]$, it is known that (LRD$_\lambda$) has a unique optimal solution [24]. We can now write the KKT conditions of problems (LRP$_\lambda$) and (LRD$_\lambda$) as

$$\langle \theta_\lambda^*, \bar{\mathbf{x}}^j \rangle \in \begin{cases} m\lambda, & \text{if } [\beta_\lambda^*]_j > 0, \\ -m\lambda, & \text{if } [\beta_\lambda^*]_j < 0, \quad j = 1, \dots, p. \\ [-m\lambda, m\lambda], & \text{if } [\beta_\lambda^*]_j = 0. \end{cases} \qquad (3)$$

In view of Eq. (3), we can see that

$$|\langle \theta_\lambda^*, \bar{\mathbf{x}}^j \rangle| < m\lambda \Rightarrow [\beta_\lambda^*]_j = 0. \qquad \text{(R1)}$$

In other words, if $|\langle \theta_\lambda^*, \bar{\mathbf{x}}^j \rangle| < m\lambda$, then the KKT conditions imply that the coefficient of $\bar{\mathbf{x}}^j$ in the solution $\beta_\lambda^*$ is 0 and thus the $j^{th}$ feature can be safely removed from the optimization of (LRP$_\lambda$). However, for the general case in which $\lambda < \lambda_{max}$, (R1) is not applicable since it assumes the knowledge of $\theta_\lambda^*$. Although it is unknown, we can still estimate a region $\mathcal{A}_\lambda$ which contains $\theta_\lambda^*$. As a result, if $\max_{\theta \in \mathcal{A}_\lambda} |\langle \theta, \bar{\mathbf{x}}^j \rangle| < m\lambda$, we can also conclude that $[\beta_\lambda^*]_j = 0$ by (R1). In other words, (R1) can be relaxed as

$$T(\theta_\lambda^*, \bar{\mathbf{x}}^j) := \max_{\theta \in \mathcal{A}_\lambda} |\langle \theta, \bar{\mathbf{x}}^j \rangle| < m\lambda \Rightarrow [\beta_\lambda^*]_j = 0. \qquad \text{(R1}'\text{)}$$

In this paper, (R1$'$) serves as the foundation for constructing our screening rules, Slores. From (R1$'$), it is easy to see that screening rules with smaller $T(\theta_\lambda^*, \bar{\mathbf{x}}^j)$ are more aggressive in discarding features. To give a tight estimation of $T(\theta_\lambda^*, \bar{\mathbf{x}}^j)$, we need to restrict the region $\mathcal{A}_\lambda$ which includes $\theta_\lambda^*$ as small as possible. In Section 3, we show that the estimation of the upper bound $T(\theta_\lambda^*, \bar{\mathbf{x}}^j)$ can be obtained via solving a convex optimization problem. We show in Section 4 that the convex optimization problem admits a closed form solution and derive Slores in Section 5 based on (R1$'$).

## 3 Estimating the Upper Bound via Solving a Convex Optimization Problem

In this section, we present a novel framework to estimate an upper bound $T(\theta_\lambda^*, \bar{\mathbf{x}}^j)$ of $|\langle \theta_\lambda^*, \bar{\mathbf{x}}^j \rangle|$. In the subsequent development, we assume a parameter $\lambda_0$ and the corresponding dual optimal $\theta_{\lambda_0}^*$ are given. In our Slores rule to be presented in Section 5, we set $\lambda_0$ and $\theta_{\lambda_0}^*$ to be $\lambda_{max}$ and $\theta_{\lambda_{max}}^*$ given in Eqs. (1) and (2). We formulate the estimation of $T(\theta_\lambda^*, \bar{\mathbf{x}}^j)$ as a constrained convex optimization problem in this section, which will be shown to admit a closed form solution in Section 4.

For the dual function $g(\theta)$, it follows that $[\nabla g(\theta)]_i = \frac{1}{m} \log(\frac{\theta_i}{1-\theta_i})$, $[\nabla^2 g(\theta)]_{i,i} = \frac{1}{m} \frac{1}{\theta_i(1-\theta_i)} \geq \frac{4}{m}$. Since $\nabla^2 g(\theta)$ is a diagonal matrix, it follows that $\nabla^2 g(\theta) \succeq \frac{4}{m} I$, where $I$ is the identity matrix. Thus, $g(\theta)$ is strongly convex with modulus $\mu = \frac{4}{m}$ [16]. Rigorously, we have the following lemma.

**Lemma 1.** *Let $\lambda > 0$ and $\theta_1, \theta_2 \in \mathcal{F}_\lambda$, then*

a). $\qquad\qquad g(\theta_2) - g(\theta_1) \geq \langle \nabla g(\theta_1), \theta_2 - \theta_1 \rangle + \frac{2}{m}\|\theta_2 - \theta_1\|_2^2. \qquad (4)$

b). *If $\theta_1 \neq \theta_2$, the inequality in (4) becomes a strict inequality, i.e., "$\geq$" becomes "$>$".*

Given $\lambda \in (0, \lambda_0]$, it is easy to see that both of $\theta_\lambda^*$ and $\theta_{\lambda_0}^*$ belong to $\mathcal{F}_{\lambda_0}$. Therefore, Lemma 1 can be a useful tool to bound $\theta_\lambda^*$ with the knowledge of $\theta_{\lambda_0}^*$. In fact, we have the following theorem.

**Theorem 2.** *Let* $\lambda_{max} \geq \lambda_0 > \lambda > 0$*, then the following holds:*

a). 
$$\|\theta_\lambda^* - \theta_{\lambda_0}^*\|_2^2 \leq \frac{m}{2} \left[ g\left( \frac{\lambda}{\lambda_0} \theta_{\lambda_0}^* \right) - g(\theta_{\lambda_0}^*) + \left( 1 - \frac{\lambda}{\lambda_0} \right) \langle \nabla g(\theta_{\lambda_0}^*), \theta_{\lambda_0}^* \rangle \right] \tag{5}$$

b). *If* $\theta_\lambda^* \neq \theta_{\lambda_0}^*$*, the inequality in (5) becomes a strict inequality, i.e., "$\leq$" becomes "$<$".*

Theorem 2 implies that $\theta_\lambda^*$ is inside a ball centred at $\theta_{\lambda_0}^*$ with radius

$$r = \sqrt{\frac{m}{2} \left[ g\left( \frac{\lambda}{\lambda_0} \theta_{\lambda_0}^* \right) - g(\theta_{\lambda_0}^*) + (1 - \frac{\lambda}{\lambda_0}) \langle \nabla g(\theta_{\lambda_0}^*), \theta_{\lambda_0}^* \rangle \right]}. \tag{6}$$

Recall that to make our screening rules more aggressive in discarding features, we need to get a tight upper bound $T(\theta_\lambda^*, \bar{\mathbf{x}}^j)$ of $|\langle \theta_\lambda^*, \bar{\mathbf{x}}^j \rangle|$ [please see (R1')]. Thus, it is desirable to further restrict the possible region $\mathcal{A}_\lambda$ of $\theta_\lambda^*$. Clearly, we can see that

$$\langle \theta_\lambda^*, \mathbf{b} \rangle = 0 \tag{7}$$

since $\theta_\lambda^*$ is feasible for problem (LRD$_\lambda$). On the other hand, we call the set $\mathcal{I}_{\lambda_0} = \{ j : \langle \theta_{\lambda_0}^*, \bar{\mathbf{x}}^j \rangle = |m\lambda_0|, j = 1, \ldots, p \}$ the "*active set*" of $\theta_{\lambda_0}^*$. We have the following lemma for the active set.

**Lemma 3.** *Given the optimal solution* $\theta_\lambda^*$ *of problem (LRD$_\lambda$), the active set* $\mathcal{I}_\lambda = \{ j : |\langle \theta_\lambda^*, \bar{\mathbf{x}}^j \rangle| = m\lambda, j = 1, \ldots, p \}$ *is not empty if* $\lambda \in (0, \lambda_{max}]$*.*

Since $\lambda_0 \in (0, \lambda_{max}]$, we can see that $\mathcal{I}_{\lambda_0}$ is not empty by Lemma 3. We pick $j_0 \in \mathcal{I}_{\lambda_0}$ and set

$$\bar{\mathbf{x}}^* = \text{sign}(\langle \theta_{\lambda_0}^*, \bar{\mathbf{x}}^{j_0} \rangle) \bar{\mathbf{x}}^{j_0}. \tag{8}$$

It follows that $\langle \bar{\mathbf{x}}^*, \theta_{\lambda_0}^* \rangle = m\lambda_0$. Due to the feasibility of $\theta_\lambda^*$ for problem (LRD$_\lambda$), $\theta_\lambda^*$ satisfies

$$\langle \theta_\lambda^*, \bar{\mathbf{x}}^* \rangle \leq m\lambda. \tag{9}$$

As a result, Theorem 2, Eq. (7) and (9) imply that $\theta_\lambda^*$ is contained in the following set:

$$\mathcal{A}_{\lambda_0}^\lambda := \{ \theta : \|\theta - \theta_{\lambda_0}^*\|_2^2 \leq r^2, \langle \theta, \mathbf{b} \rangle = 0, \langle \theta, \bar{\mathbf{x}}^* \rangle \leq m\lambda \}.$$

Since $\theta_\lambda^* \in \mathcal{A}_{\lambda_0}^\lambda$, we can see that $|\langle \theta_\lambda^*, \bar{\mathbf{x}}^j \rangle| \leq \max_{\theta \in \mathcal{A}_{\lambda_0}^\lambda} |\langle \theta, \bar{\mathbf{x}}^j \rangle|$. Therefore, (R1') implies that if

$$T(\theta_\lambda^*, \bar{\mathbf{x}}^j; \theta_{\lambda_0}^*) := \max_{\theta \in \mathcal{A}_{\lambda_0}^\lambda} |\langle \theta, \bar{\mathbf{x}}^j \rangle| \tag{UBP}$$

is smaller than $m\lambda$, we can conclude that $[\beta_\lambda^*]_j = 0$ and $\bar{\mathbf{x}}^j$ can be discarded from the optimization of (LRP$_\lambda$). Notice that, we replace the notations $\mathcal{A}_\lambda$ and $T(\theta_\lambda^*, \bar{\mathbf{x}}^j)$ with $T(\theta_\lambda^*, \bar{\mathbf{x}}^j; \theta_{\lambda_0}^*)$ and $\mathcal{A}_{\lambda_0}^\lambda$ to emphasize their dependence on $\theta_{\lambda_0}^*$. Clearly, as long as we can solve for $T(\theta_\lambda^*, \bar{\mathbf{x}}^j; \theta_{\lambda_0}^*)$, (R1') would be an applicable screening rule to discard features which have 0 coefficients in $\beta_\lambda^*$. We give a closed form solution of problem (UBP) in the next section.

## 4  Solving the Convex Optimization Problem (UBP)

In this section, we show how to solve the convex optimization problem (UBP) based on the standard Lagrangian multiplier method. We first transform problem (UBP) into a pair of convex minimization problem (UBP') via Eq. (11) and then show that the strong duality holds for (UBP') in Lemma 6. The strong duality guarantees the applicability of the Lagrangian multiplier method. We then give the closed form solution of (UBP') in Theorem 8. After we solve problem (UBP'), it is straightforward to compute the solution of problem (UBP) via Eq. (11).

Before we solve (UBP) for the general case, it is worthwhile to mention a special case in which $\mathbf{P}\bar{\mathbf{x}}^j = \bar{\mathbf{x}}^j - \frac{\langle \bar{\mathbf{x}}^j, \mathbf{b} \rangle}{\|\mathbf{b}\|_2^2} \mathbf{b} = 0$. Clearly, $\mathbf{P}$ is the projection operator which projects a vector onto the orthogonal complement of the space spanned by $\mathbf{b}$. In fact, we have the following theorem.

**Theorem 4.** *Let* $\lambda_{max} \geq \lambda_0 > \lambda > 0$*, and assume* $\theta_{\lambda_0}^*$ *is known. For* $j \in \{1, \ldots, p\}$*, if* $\mathbf{P}\bar{\mathbf{x}}^j = 0$*, then* $T(\theta_\lambda^*, \bar{\mathbf{x}}^j; \theta_{\lambda_0}^*) = 0$*.*

Because of (R1′), we immediately have the following corollary.

**Corollary 5.** *Let $\lambda \in (0, \lambda_{max})$ and $j \in \{1, \ldots, p\}$. If $\mathbf{P}\bar{\mathbf{x}}^j = 0$, then $[\beta_\lambda^*]_j = 0$.*

For the general case in which $\mathbf{P}\bar{\mathbf{x}}^j \neq 0$, let

$$T_+(\theta_\lambda^*, \bar{\mathbf{x}}^j; \theta_{\lambda_0}^*) := \max_{\theta \in \mathcal{A}_{\lambda_0}^\lambda} \langle \theta, +\bar{\mathbf{x}}^j \rangle, \ T_-(\theta_\lambda^*, \bar{\mathbf{x}}^j; \theta_{\lambda_0}^*) := \max_{\theta \in \mathcal{A}_{\lambda_0}^\lambda} \langle \theta, -\bar{\mathbf{x}}^j \rangle. \tag{10}$$

Clearly, we have

$$T(\theta_\lambda^*, \bar{\mathbf{x}}^j; \theta_{\lambda_0}^*) = \max\{T_+(\theta_\lambda^*, \bar{\mathbf{x}}^j; \theta_{\lambda_0}^*), T_-(\theta_\lambda^*, \bar{\mathbf{x}}^j; \theta_{\lambda_0}^*)\}. \tag{11}$$

Therefore, we can solve problem (UBP) by solving the two sub-problems in (10).

Let $\xi \in \{+1, -1\}$. Then problems in (10) can be written uniformly as

$$T_\xi(\theta_\lambda^*, \bar{\mathbf{x}}^j; \theta_{\lambda_0}^*) = \max_{\theta \in \mathcal{A}_{\lambda_0}^\lambda} \langle \theta, \xi\bar{\mathbf{x}}^j \rangle. \tag{UBP$_s$}$$

To make use of the standard Lagrangian multiplier method, we transform problem (UBP$_s$) to the following minimization problem:

$$-T_\xi(\theta_\lambda^*, \bar{\mathbf{x}}^j; \theta_{\lambda_0}^*) = \min_{\theta \in \mathcal{A}_{\lambda_0}^\lambda} \langle \theta, -\xi\bar{\mathbf{x}}^j \rangle \tag{UBP′}$$

by noting that $\max_{\theta \in \mathcal{A}_{\lambda_0}^\lambda} \langle \theta, \xi\bar{\mathbf{x}}^j \rangle = -\min_{\theta \in \mathcal{A}_{\lambda_0}^\lambda} \langle \theta, -\xi\bar{\mathbf{x}}^j \rangle$.

**Lemma 6.** *Let $\lambda_{max} \geq \lambda_0 > \lambda > 0$ and assume $\theta_{\lambda_0}^*$ is known. The strong duality holds for problem (UBP′). Moreover, problem (UBP′) admits an optimal solution in $\mathcal{A}_{\lambda_0}^\lambda$.*

Because the strong duality holds for problem (UBP′) by Lemma 6, the Lagrangian multiplier method is applicable for (UBP′). In general, we need to first solve the dual problem and then recover the optimal solution of the primal problem via KKT conditions. Recall that $r$ and $\bar{\mathbf{x}}^*$ are defined by Eq. (6) and (8) respectively. Lemma 7 derives the dual problems of (UBP′) for different cases.

**Lemma 7.** *Let $\lambda_{max} \geq \lambda_0 > \lambda > 0$ and assume $\theta_{\lambda_0}^*$ is known. For $j \in \{1, \ldots, p\}$ and $\mathbf{P}\bar{\mathbf{x}}^j \neq 0$, let $\bar{\mathbf{x}} = -\xi\bar{\mathbf{x}}^j$. Denote*

$$\mathcal{U}_1 = \{(u_1, u_2) : u_1 > 0, u_2 \geq 0\} \text{ and } \mathcal{U}_2 = \left\{(u_1, u_2) : u_1 = 0, u_2 = -\frac{\langle \mathbf{P}\bar{\mathbf{x}}, \mathbf{P}\bar{x}^* \rangle}{\|\mathbf{P}\bar{\mathbf{x}}^*\|_2^2}\right\}.$$

*a). If $\frac{\langle \mathbf{P}\bar{\mathbf{x}}, \mathbf{P}\bar{x}^* \rangle}{\|\mathbf{P}\bar{\mathbf{x}}\|_2 \|\mathbf{P}\bar{x}^*\|_2} \in (-1, 1]$, the dual problem of (UBP′) is equivalent to:*

$$\max_{(u_1, u_2) \in \mathcal{U}_1} \bar{g}(u_1, u_2) = -\frac{1}{2u_1}\|\mathbf{P}\bar{\mathbf{x}} + u_2\mathbf{P}\bar{\mathbf{x}}^*\|_2^2 + u_2 m(\lambda_0 - \lambda) + \langle \theta_{\lambda_0}^*, \bar{\mathbf{x}} \rangle - \frac{1}{2}u_1 r^2. \tag{UBD′}$$

*Moreover, $\bar{g}(u_1, u_2)$ attains its maximum in $\mathcal{U}_1$.*

*b). If $\frac{\langle \mathbf{P}\bar{\mathbf{x}}, \mathbf{P}\bar{x}^* \rangle}{\|\mathbf{P}\bar{\mathbf{x}}\|_2 \|\mathbf{P}\bar{x}^*\|_2} = -1$, the dual problem of (UBP′) is equivalent to:*

$$\max_{(u_1, u_2) \in \mathcal{U}_1 \cup \mathcal{U}_2} \bar{\bar{g}}(u_1, u_2) = \begin{cases} \bar{g}(u_1, u_2), & \text{if } (u_1, u_2) \in \mathcal{U}_1, \\ -\frac{\|\mathbf{P}\bar{\mathbf{x}}\|_2}{\|\mathbf{P}\bar{\mathbf{x}}^*\|_2}m\lambda, & \text{if } (u_1, u_2) \in \mathcal{U}_2. \end{cases} \tag{UBD″}$$

We can now solve problem (UBP′) in the following theorem.

**Theorem 8.** *Let $\lambda_{max} \geq \lambda_0 > \lambda > 0$, $d = \frac{m(\lambda_0 - \lambda)}{r\|\mathbf{P}\bar{x}^*\|_2}$ and assume $\theta_{\lambda_0}^*$ is known. For $j \in \{1, \ldots, p\}$ and $\mathbf{P}\bar{\mathbf{x}}^j \neq 0$, let $\bar{\mathbf{x}} = -\xi\bar{\mathbf{x}}^j$.*

*a). If $\frac{\langle \mathbf{P}\bar{\mathbf{x}}, \mathbf{P}\bar{x}^* \rangle}{\|\mathbf{P}\bar{\mathbf{x}}\|_2 \|\mathbf{P}\bar{x}^*\|_2} \geq d$, then*

$$T_\xi(\theta_\lambda^*, \bar{\mathbf{x}}^j; \theta_{\lambda_0}^*) = r\|\mathbf{P}\bar{\mathbf{x}}\|_2 - \langle \theta_{\lambda_0}^*, \bar{\mathbf{x}} \rangle; \tag{12}$$

b). *If* $\frac{\langle \mathbf{P}\bar{\mathbf{x}}, \mathbf{P}\bar{\mathbf{x}}^* \rangle}{\|\mathbf{P}\bar{\mathbf{x}}\|_2 \|\mathbf{P}\bar{x}^*\|_2} < d$, *then*

$$T_\xi(\theta_\lambda^*, \bar{\mathbf{x}}^j; \theta_{\lambda_0}^*) = r\|\mathbf{P}\bar{\mathbf{x}} + u_2^* \mathbf{P}\bar{\mathbf{x}}^*\|_2 - u_2^* m(\lambda_0 - \lambda) - \langle \theta_{\lambda_0}^*, \bar{\mathbf{x}} \rangle, \tag{13}$$

*where*

$$
\begin{aligned}
u_2^* &= \frac{-a_1 + \sqrt{\Delta}}{2a_2}, \\
a_2 &= \|\mathbf{P}\bar{\mathbf{x}}^*\|_2^4 (1 - d^2), \\
a_1 &= 2\langle \mathbf{P}\bar{\mathbf{x}}, \mathbf{P}\bar{\mathbf{x}}^* \rangle \|\mathbf{P}\bar{\mathbf{x}}^*\|_2^2 (1 - d^2), \\
a_0 &= \langle \mathbf{P}\bar{\mathbf{x}}, \mathbf{P}\bar{\mathbf{x}}^* \rangle^2 - d^2 \|\mathbf{P}\bar{\mathbf{x}}\|_2^2 \|\mathbf{P}\bar{\mathbf{x}}^*\|_2^2, \\
\Delta &= a_1^2 - 4a_2 a_0 = 4d^2(1 - d^2)\|\mathbf{P}\bar{\mathbf{x}}^*\|_2^4(\|\mathbf{P}\bar{\mathbf{x}}\|_2^2 \|\mathbf{P}\bar{\mathbf{x}}^*\|_2^2 - \langle \mathbf{P}\bar{\mathbf{x}}, \mathbf{P}\bar{\mathbf{x}}^* \rangle^2).
\end{aligned}
\tag{14}
$$

Notice that, although the dual problems of (UBP$'$) in Lemma 7 are different, the resulting upper bound $T_\xi(\theta_\lambda^*, \bar{\mathbf{x}}^j; \theta_{\lambda_0}^*)$ can be given by Theorem 8 in a uniform way. The tricky part is how to deal with the extremal cases in which $\frac{\langle \mathbf{P}\bar{\mathbf{x}}, \mathbf{P}\bar{\mathbf{x}}^* \rangle}{\|\mathbf{P}\bar{\mathbf{x}}\|_2 \|\mathbf{P}\bar{x}^*\|_2} \in \{-1, +1\}$.

## 5  The proposed Slores Rule for $\ell_1$ Regularized Logistic Regression

Using (R1$'$), we are now ready to construct the screening rules for the $\ell_1$ Regularized Logistic Regression. By Corollary 5, we can see that the orthogonality between the $j^{th}$ feature and the response vector $\mathbf{b}$ implies the absence of $\bar{\mathbf{x}}^j$ from the resulting model. For the general case in which $\mathbf{P}\bar{\mathbf{x}}^j \neq 0$, (R1$'$) implies that if $T(\theta_\lambda^*, \bar{\mathbf{x}}^j; \theta_{\lambda_0}^*) = \max\{T_+(\theta_\lambda^*, \bar{\mathbf{x}}^j; \theta_{\lambda_0}^*), T_-(\theta_\lambda^*, \bar{\mathbf{x}}^j; \theta_{\lambda_0}^*)\} < m\lambda$, then the $j^{th}$ feature can be discarded from the optimization of (LRP$_\lambda$). Notice that, letting $\xi = \pm 1$, $T_+(\theta_\lambda^*, \bar{\mathbf{x}}^j; \theta_{\lambda_0}^*)$ and $T_-(\theta_\lambda^*, \bar{\mathbf{x}}^j; \theta_{\lambda_0}^*)$ have been solved by Theorem 8. Rigorously, we have the following theorem.

**Theorem 9** (Slores). *Let $\lambda_0 > \lambda > 0$ and assume $\theta_{\lambda_0}^*$ is known.*

1. *If $\lambda \geq \lambda_{max}$, then $\beta_\lambda^* = 0$;*
2. *If $\lambda_{max} \geq \lambda_0 > \lambda > 0$ and either of the following holds:*
   *(a) $\mathbf{P}\bar{\mathbf{x}}^j = 0$,*
   *(b) $\max\{T_\xi(\theta_\lambda^*, \bar{\mathbf{x}}^j; \theta_{\lambda_0}^*) : \xi = \pm 1\} < m\lambda$,*
   *then $[\beta_\lambda^*]_j = 0$.*

Based on Theorem 9, we construct the Slores rule as summarized below in Algorithm 1. Notice that, the output $\mathcal{R}$ of Slores is the indices of the features that need to be entered to the optimization. As a result, suppose the output of Algorithm 1 is $\mathcal{R} = \{j_1, \ldots, j_k\}$, we can substitute the full matrix $\overline{\mathbf{X}}$ in problem (LRP$_\lambda$) with the sub-matrix $\overline{\mathbf{X}}_\mathcal{R} = (\bar{\mathbf{x}}^{j_1}, \ldots, \bar{\mathbf{x}}^{j_k})$ and just solve for $[\beta_\lambda^*]_\mathcal{R}$ and $c_\lambda^*$.

On the other hand, Algorithm 1 implies that Slores needs five inputs. Since $\overline{\mathbf{X}}$ and $\mathbf{b}$ come with the data and $\lambda$ is chosen by the user, we only need to specify $\theta_{\lambda_0}^*$ and $\lambda_0$. In other words, we need to provide Slores with a dual optimal solution of problem (LRD$_\lambda$) for an arbitrary parameter. A natural choice is by setting $\lambda_0 = \lambda_{max}$ and $\theta_{\lambda_0}^* = \theta_{\lambda_{max}}^*$ given by Eq. (1) and Eq. (2) in closed form.

---

**Algorithm 1** $\mathcal{R} = \text{Slores}(\overline{\mathbf{X}}, \mathbf{b}, \lambda, \lambda_0, \theta_{\lambda_0}^*)$

---

**Initialize** $\mathcal{R} := \{1, \ldots, p\}$;
**if** $\lambda \geq \lambda_{max}$ **then**
   set $\mathcal{R} = \emptyset$;
**else**
   **for** $j = 1$ **to** $p$ **do**
      **if** $\mathbf{P}\bar{\mathbf{x}}^j = 0$ **then**
         remove $j$ from $\mathcal{R}$;
      **else if** $\max\{T_\xi(\theta_\lambda^*, \bar{\mathbf{x}}^j; \theta_{\lambda_0}^*) : \xi = \pm 1\} < m\lambda$
      **then**
         remove $j$ from $\mathcal{R}$;
      **end if**
   **end for**
**end if**
**Return:** $\mathcal{R}$

---

## 6  Experiments

We evaluate our screening rules using the newgroup data set [10] and Yahoo web pages data sets [23]. The newgroup data set is cultured from the data by Koh et al. [10]. The Yahoo data sets include 11 top-level categories, each of which is further divided into a set of subcategories. In

our experiment we construct five balanced binary classification datasets from the topics of Computers, Education, Health, Recreation, and Science. For each topic, we choose samples from one subcategory as the positive class and randomly sample an equal number of samples from the rest of subcategories as the negative class. The statistics of the data sets are given in Table 1.

We compare the performance of Slores and the strong rule which achieves state-of-the-art performance for $\ell_1$ regularized LR. We do not include SAFE because it is less effective in discarding features than strong rules and requires much higher computational time [22]. Fig. 1 has shown the performance of Slores, strong rule and SAFE. We compare the efficiency of the three screening rules on the same prostate cancer data set in Table 2. All of the screening rules are tested along a sequence of 86 parameter values equally spaced on the $\lambda/\lambda_{max}$ scale from 0.1 to 0.95. We repeat the procedure 100 times and during each time we undersam-

Table 1: Statistics of the test data sets.

| Data set | $m$ | $p$ | no. nonzeros |
|---|---|---|---|
| newsgroup | 11269 | 61188 | 1467345 |
| Computers | 216 | 25259 | 23181 |
| Education | 254 | 20782 | 28287 |
| Health | 228 | 18430 | 40145 |
| Recreation | 370 | 25095 | 49986 |
| Science | 222 | 24002 | 37227 |

Table 2: Running time (in seconds) of Slores, strong rule, SAFE and the solver.

| Slores | Strong Rule | SAFE | Solver |
|---|---|---|---|
| 0.37 | 0.33 | 1128.65 | 10.56 |

ple 80% of the data. We report the total running time of the three screening rules over the 86 values of $\lambda/\lambda_{max}$ in Table 2. For reference, we also report the total running time of the solver[1]. We observe that the running time of Slores and strong rule is negligible compared to that of the solver. However, SAFE takes much longer time even than the solver.

In Section 6.1, we evaluate the performance of Slores and strong rule. Recall that we use the rejection ratio, i.e., the ratio between the number of features discarded by the screening rules and the number of features with 0 coefficients in the solution, to measure the performance of screening rules. Note that, because no features with non-zero coefficients in the solution would be mistakenly discarded by Slores, its rejection ratio is no larger than one. We then compare the efficiency of Slores and strong rule in Section 6.2.

The experiment settings are as follows. For each data set, we undersample 80% of the date and run Slores and strong rules along a sequence of 86 parameter values equally spaced on the $\lambda/\lambda_{max}$ scale from 0.1 to 0.95. We repeat the procedure 100 times and report the average performance and running time at each of the 86 values of $\lambda/\lambda_{max}$. Slores, strong rules and SAFE are all implemented in Matlab. All of the experiments are carried out on a Intel(R) (i7-2600) 3.4Ghz processor.

### 6.1 Comparison of Performance

In this experiment, we evaluate the performance of the Slores and the strong rule via the rejection ratio. Fig. 2 shows the rejection ratio of Slores and strong rule on six real data sets. When $\lambda/\lambda_{max} > 0.5$, we can see that both Slores and strong rule are able to identify almost 100% of the inactive features, i.e., features with 0 coefficients in the solution vector. However, when $\lambda/\lambda_{max} \leq 0.5$, strong rule can not detect the inactive features. In contrast, we observe that Slores exhibits much stronger capability in discarding inactive features for small $\lambda$, even when $\lambda/\lambda_{max}$ is close to 0.1. Taking the data point at which $\lambda/\lambda_{max} = 0.1$ for example, Slores discards about 99% inactive features for the newsgroup data set. For the other data sets, more than 80% inactive features are identified by Slores. Thus, in terms of rejection ratio, Slores significantly outperforms the strong rule. Moreover, the discarded features by Slores are guaranteed to have 0 coefficients in the solution. But strong rule may mistakenly discard features which have non-zero coefficients in the solution.

### 6.2 Comparison of Efficiency

We compare efficiency of Slores and the strong rule in this experiment. The data sets for evaluating the rules are the same as Section 6.1. The running time of the screening rules reported in Fig. 3 includes the computational cost of the rules themselves and that of the solver after screening. We plot the running time of the screening rules against that of the solver without screening. As indicated by Fig. 2, when $\lambda/\lambda_{max} > 0.5$, Slores and strong rule discards almost 100% of the inactive features.

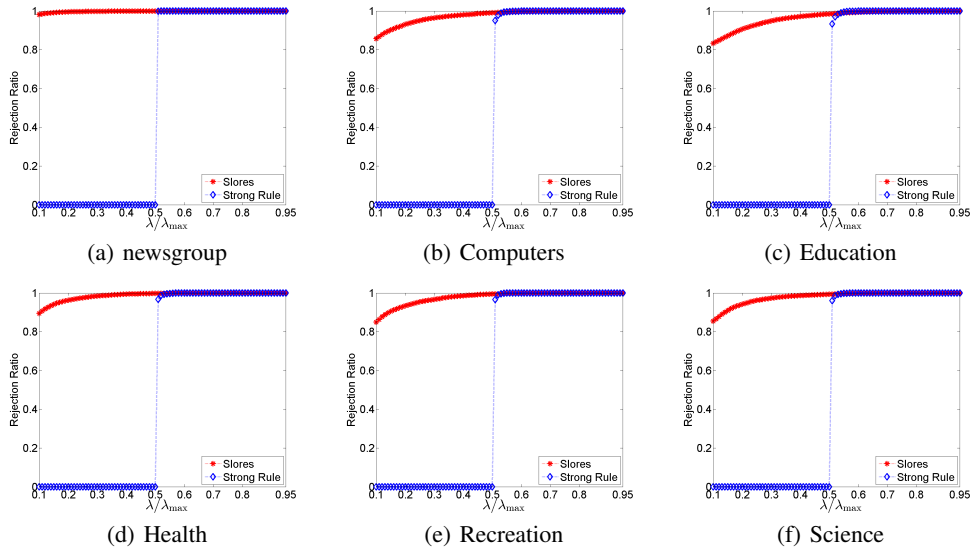

Figure 2: Comparison of the performance of Slores and strong rules on six real data sets.

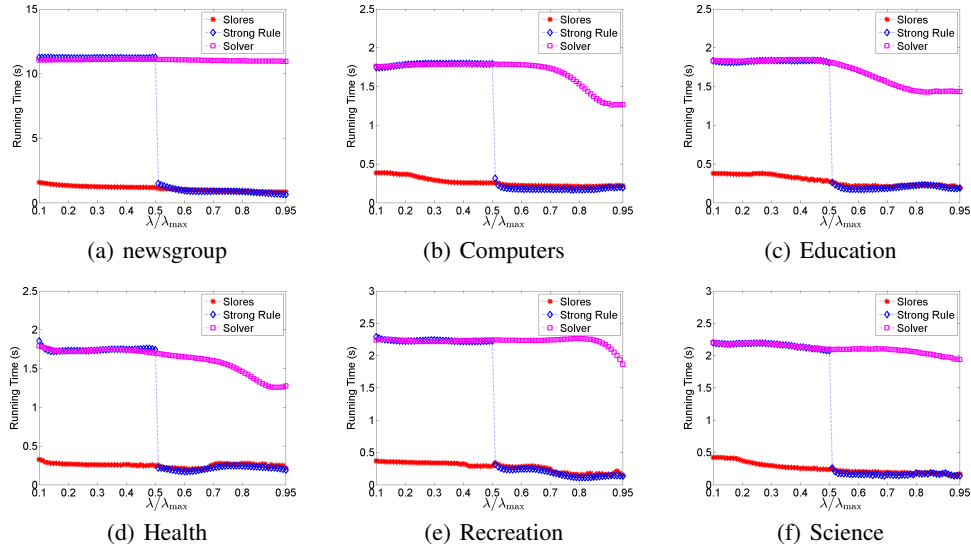

Figure 3: Comparison of the efficiency of Slores and strong rule on six real data sets.

As a result, the size of the feature matrix involved in the optimization of problem ($\text{LRP}_\lambda$) is greatly reduced. From Fig. 3, we can observe that the efficiency is improved by about one magnitude on average compared to that of the solver without screening. However, when $\lambda/\lambda_{max} < 0.5$, strong rule can not identify any inactive features and thus the running time is almost the same as that of the solver without screening. In contrast, Slores is still able to identify more than $80\%$ of the inactive features for the data sets cultured from the Yahoo web pages data sets and thus the efficiency is improved by roughly 5 times. For the newgroup data set, about $99\%$ inactive features are identified by Slores which leads to about 10 times savings in running time. These results demonstrate the power of the proposed Slores rule in improving the efficiency of solving the $\ell_1$ regularized LR.

## 7 Conclusions

In this paper, we propose novel screening rules to effectively discard features for $\ell_1$ regularized LR. Extensive numerical experiments on real data demonstrate that Slores outperforms the existing state-of-the-art screening rules. We plan to extend the framework of Slores to more general sparse formulations, including convex ones, like group Lasso, fused Lasso, $\ell_1$ regularized SVM, and non-convex ones, like $\ell_p$ regularized problems where $0 < p < 1$.

## Footnotes

[1]In this paper, the ground truth is computed by SLEP [14].

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
