[Reviews · NeurIPS 2014]

Submitted by Assigned_Reviewer_2

The authors propose a new screening rule Slores for pre-filtering variables for logistic regression. This statement though sounds too simple and doesn't give the paper justice at all. The paper provides a rigorous and theoretically well founded derivation of a novel pre-screening rule which could in principle be extended to other settings as well. The method is also efficient compared to other safe rules that guarantee to discard only non-zero entries. The experiments compare 3 algorithms, and a solver of Lasso, here SLEP (all implemented in Matlab, for the same 2 data sources).
The paper is substantially novel providing a technical as well as practical contribution. The paper is well written. There are just a few typos (that I have noticed, there maybe more):

P3 just after R1 the absolute value is not complete
Supplementary: page 2 : typo in the equation just before (7) lambda seems to be missing
Summary: The paper is substantially novel providing theoretical, technical as well as practical contribution. The paper is well written.

Submitted by Assigned_Reviewer_7

In this paper, the authors propose a new sparse logistic regression screening rule (Slores) for feature selection, which discard the features with zero coefficients by solving a convex optimization problem with closed form solution. A series of evaluations have been conducted to show its effectiveness.

This idea is simple but interesting and I like it. However, I still have some questions:
1. In the section 3, the authors propose to estimate the upper bound by solving a optimization problem, where the critical parameter $\lambda_0$ and $\theta_{\lambda_0}^\ast$ are set to be $\lambda_\max$ and $\theta_{\lambda_{max}}^\ast$ respectively. However, how can we know $\lambda_\max$ and $\theta_{\lambda_{max}}^\ast$ at the beginning?
2. In the 138th row, a “|” is missing in the equation.
3. In the experimental results, the authors just use the reject ratio to measure the performance. However, I still have a problem: whether the non-zero coefficients can be affected by discarding these features? Some other feature selection measures are recommended.
4. The authors should compare the proposed method with the state-of-the-art, such as “ Lasso Screening Rules via Dual Polytope Projection, ICML, 2013”.

Summary: In this paper, the authors propose a new sparse logistic regression screening rule (Slores) for feature selection, which discard the features with zero coefficients by solving a convex optimization problem with closed form solution.
This idea is simple but interesting and I like it.

Submitted by Assigned_Reviewer_12

In this paper, a new screening rules for $l_1$ regularized LR is presented, called “Slores”. The experimental results show that Slores gives better performance than existing screening rules and solves SLR efficiently.

However, this paper involves a lot of mathematic derivations that are very difficult to understand and many lemma and theorems are proposed with their motivations unclear. The readability of paper is very poor for real applications.

The experimental comparisons are insufficient. There are many other feature selection approaches, but the authors only compared Slores with strong rules. Hence, more experiments are expected. Also the descriptions and discussions of experiments are not clear and it is difficult for readers to implement the proposed method.

Summary: This paper gives a new screening rules for $l_1$ regularized LR called “Slores”. The theoretical derivations are not well explained and difficult to understand. Experiments are not sufficient.
Author Feedback
Author rebuttal: We thank all reviewers for the constructive comments.

Reviewer 12

Q: The theoretical deviations are difficult to understand and the motivations of some theorems and lemmas are not well explained.

A: Thanks for pointing this out. Most existing works on safe screening are focused on the Lasso problem. To our best knowledge, there are very few safe screening algorithms that are applicable for sparse logistic regression, which is the focus of the current paper. Our paper shows for the first time in the literature that effective safe screening for sparse logistic regression is possible, however the deviation is quite complicated and highly nontrivial. This is the major contribution of this paper. We will try to improve the readability of the final version, if accepted.

The theoretical deviations are mostly based on convex optimization. Many efforts are devoted to make this submission theoretically rigorous. The main results in this paper are Theorems 2, 8 and 9. The most difficult part is to show Theorem 8, i.e., solve the constrained convex optimization problem (UBP). Theorem 2(a) is to construct the feasible region of (UBP). Theorem 2(b) is needed to verify the Slater’s condition such that strong duality holds for (UBP) [please refer to Lemma 6)]. Thus, we can solve (UBP) via its dual problem, which is given by Lemma 7. The tricky part is due to the degenerate case [Lemma 7(b)].

Q: More experiments are expected to compare other feature selection approaches.

A: This may be a misunderstanding. In this paper, we do not aim to develop a new feature selection method. Our goal is to develop a novel tool, called Slores screening rule, to accelerate the computation of spare logistic regression. As another reviewer pointed out, Slores aims to quickly pre-filter the zero components in the solution vector of sparse logistic regression model. As a result, the number of features to be entered to the optimization could be greatly reduced, which may lead to substantial savings in the memory usage and computational cost.

Reviewer 2

Q: The paper is substantially novel providing theoretical, technical as well as practical contribution. There are just a few typos.

A: Thanks for the compliment. We will correct all of the typos accordingly in the final version, if accepted.

Reviewer 7

Q: How do we know lambda_max and theta_(lambda_max) at the beginning?

A: lambda_max and theta_(lambda_max) are given by Eqs (1) and (2), respectively. Please refer to [10] for detail.

Q: In the 138th row, a “|” is missing in the equation.

A: Thanks for pointing this out. We will correct all of the typos accordingly in the final version, if accepted.

Q: Can the non-zero coefficients be affected by discarding these features?

A: The discarded features are guaranteed to have zero coefficients. Therefore, solving the sparse logistic regression problem on the reduced data set will not change the non-zero coefficients of the remaining features.

Q: Other feature selection measures are recommended.

A: Thanks for this nice suggestion. The rejection ratio is the standard measure of the performance of the screening methods in the existing literature, e.g., [6,23,26,27]. Besides the rejection ratio, we also compare the running time of the solver without screening and the solver with Slores (please refer to Fig. 3). The results indicate that the speedup gained by Slores is substantial.

Q: The author should compare the state-of-the-art, such as “Lasso Screening Rules via Dual Polytope Projection, NIPS2013”.

A: Slores developed in this paper is for sparse logistic regression. However, the DPP rule is developed for the standard Lasso problem and it is not applicable for sparse logistic regression. We would like to emphasize that most existing works on safe screening, e.g., DPP are focused on the Lasso problem.